# Defective Biomechanics and Pharmacological Rescue of Human Cardiomyocytes with Filamin C Truncations

**DOI:** 10.3390/ijms25052942

**Published:** 2024-03-03

**Authors:** Marco Lazzarino, Michele Zanetti, Suet Nee Chen, Shanshan Gao, Brisa Peña, Chi Keung Lam, Joseph C. Wu, Matthew R. G. Taylor, Luisa Mestroni, Orfeo Sbaizero

**Affiliations:** 1CNR-IOM, Area Science Park, 34149 Trieste, Italy; lazzarino@iom.cnr.it (M.L.); zanetti@iom.cnr.it (M.Z.); 2Cardiovascular Institute, University of Colorado Anschutz Medical Campus, Aurora, CO 80045, USA; suet.chen@cuanschutz.edu (S.N.C.); shanshan.gao@cuanschutz.edu (S.G.); brisa.penacastellanos@cuanschutz.edu (B.P.); matthew.taylor@cuanschutz.edu (M.R.G.T.); luisa.mestroni@cuanschutz.edu (L.M.); 3Bioengineering Department, University of Colorado Anschutz Medical Campus, Aurora, CO 80045, USA; 4Stanford Cardiovascular Institute, Stanford University, Stanford, CA 94305, USA; lamcg@udel.edu (C.K.L.); joewu@stanford.edu (J.C.W.); 5Engineering and Architecture Department, University of Trieste, 34127 Trieste, Italy

**Keywords:** filamin C, atomic force microscopy, induced pluripotent stem cells, cardiomyocytes, mechanomics, arrhythmogenic cardiomyopathy, genetics

## Abstract

Actin-binding filamin C (FLNC) is expressed in cardiomyocytes, where it localizes to Z-discs, sarcolemma, and intercalated discs. Although FLNC truncation variants (*FLNCtv*) are an established cause of arrhythmias and heart failure, changes in biomechanical properties of cardiomyocytes are mostly unknown. Thus, we investigated the mechanical properties of human-induced pluripotent stem cells-derived cardiomyocytes (hiPSC-CMs) carrying *FLNCtv*. CRISPR/Cas9 genome-edited homozygous FLNC^KO−/−^ hiPSC-CMs and heterozygous knock-out FLNC^KO+/−^ hiPSC-CMs were analyzed and compared to wild-type FLNC (FLNC^WT^) hiPSC-CMs. Atomic force microscopy (AFM) was used to perform micro-indentation to evaluate passive and dynamic mechanical properties. A qualitative analysis of the beating traces showed gene dosage-dependent-manner “irregular” peak profiles in FLNC^KO+/−^ and FLNC^KO−/−^ hiPSC-CMs. Two Young’s moduli were calculated: E1, reflecting the compression of the plasma membrane and actin cortex, and E2, including the whole cell with a cytoskeleton and nucleus. Both E1 and E2 showed decreased stiffness in mutant FLNC^KO+/−^ and FLNC^KO−/−^ iPSC-CMs compared to that in FLNC^WT^. The cell adhesion force and work of adhesion were assessed using the retraction curve of the SCFS. Mutant FLNC iPSC-CMs showed gene dosage-dependent decreases in the work of adhesion and adhesion forces from the heterozygous FLNC^KO+/−^ to the FLNC^KO−/−^ model compared to FLNC^WT^, suggesting damaged cytoskeleton and membrane structures. Finally, we investigated the effect of crenolanib on the mechanical properties of hiPSC-CMs. Crenolanib is an inhibitor of the Platelet-Derived Growth Factor Receptor α (PDGFRA) pathway which is upregulated in *FLNCtv* hiPSC-CMs. Crenolanib was able to partially rescue the stiffness of FLNC^KO−/−^ hiPSC-CMs compared to control, supporting its potential therapeutic role.

## 1. Introduction

In order to react to forces, cardiomyocytes need an efficient *mechanome* [1], a mechanosensitive chain of elements spanning from the intracellular space to the extracellular matrix that provide the appropriate mechanical response to the environment. One of the most important mechanosensitive elements is the cytoskeleton, an organized network of various biological polymers. The three key components of the cytoskeleton are actin, microtubules, and intermediate filaments (that consist mainly of desmin) [2]: actin is probably the most critical, due, in part, to its highly dynamic nature and its role in a variety of cellular processes, such as cell signaling, the organization and conservation of both cell junctions and cell shape, and muscle contraction [3]. The mechanical properties of the actin network are maintained by actin cross-linking or actin-binding proteins (ABPs). 

Among ABPs, filamins (FLNs) have a unique microstructure and actin crosslinking properties. FLNs are elongated, V-shaped dimers that dimerize at their C-termini [4]. FLNs consist of an N-terminal spectrin-related ABD—called the “head”—followed by 24 immunoglobulin-like repeats (R), called the “backbone”; the 24th repeat domain is located at the C-terminus [4,5] (Figure 1A) [6]. The filamin “head” is similar to that of alpha-actinin and dystrophin, and its role is the same [4]. However, in the “backbone” there is two hinges that separate R15 from R16 (hinge-1) and R23 from R24 (hinge-2), and it confers more flexibility to the structure [5]. This geometry and flexibility allows for the FLNs to crosslink actin filaments into orthogonal networks (or in any case, it promotes the high-angle branching of actin filaments) (Figure 1B) [7].

Besides its shape, another peculiarity differentiates FLNs from the others ABPs. Contrary to inorganic materials that do not couple strength and toughness, and are either strong with slight ductility (e.g., ceramics, diamond), or quite weak with high toughness (e.g., metals like aluminum) [8,9], protein materials are capable of displaying, at the same time, properties such as strength and a high deformation capacity without fracturing despite the presence of “defects” (toughness). This ability is due to the three molecular building blocks that are predominant in all structural proteins: α-helices, β-sheets, and tropocollagen. In the case of FLNs, the rod domains are β-sheets: they show a high strength, offering resistance when loaded, approximately 10 times higher than the maximum force in α-helices. However, β-sheets bare only small strains: therefore, β-sheets fail in a more brittle way [8,9,10]. From an engineering point of view, β-sheets resemble a rock climber’s progressive-tearing energy absorber. Furthermore, even though there is no evidence that filamin and alpha-actinin interact directly, these two major actin cross-linking proteins display synergistic mechanical functions [11].

FLNs not only regulate the organization of the actin network, but they also provide a mechanical link between the extracellular matrix (ECM) and the actin cytoskeleton. In this way, FLNs keep a cell’s mechanical stability by linking the internal actin network with membrane receptors and mechanosensitive components [12]. Based on the multiple roles played by ABPs, it is not surprising that many mutations in FLNs genes, either small deletions or insertions, or truncating nonsense mutations and missense, have pathogenic consequences in the nervous system, cardiovascular system, muscles, and connective tissues [13,14,15,16,17].

FLNC is the main isoform expressed in striated muscle (cardiac and skeletal myocytes) [18,19,20]. FLNC localizes to Z-discs, sarcolemma, and intercalated discs [21,22]. As a consequence, missense and loss-of-function mutations in the *FLNC* gene have been associated with a variety of cardiomyopathies, including dilated, hypertrophic and restrictive cardiomyopathies [23,24,25,26], and myopathies [14,27,28]. Notably, while missense mutations appear more prominent in restrictive cardiomyopathy leading to abnormal protein aggregation in RCM patients [29], *FLNC* truncation variants (*FLNCtv*) are a main cause of arrhythmogenic cardiomyopathy (ACM), a disease that leads to progressive heart failure and lethal ventricular arrhythmias in humans [25,30,31,32].

*FLNC* truncating loss-of-function mutation (*FLNCtv*) are found in 2% of patients with dilated cardiomyopathy and 6% of those with an arrhythmogenic phenotype [31,32]. In our previous investigations, *FLNCtv* carriers presented with heart failure and life-threatening ventricular arrhythmias in early stages of disease [31,32,33] and no signs of muscle involvement or cytoplasmic aggregates. They showed irregular Z-line and desmosomes, localization of FLNC to the intercalated disk, and myocardial fibrosis with fatty infiltration in the left ventricle resembling arrhythmogenic right ventricular cardiomyopathy and providing the substrate for arrhythmias. Immunofluorescence studies showed reduced signals for desmoplakin and SAP97 at the intercalated disks, both in the cardiac tissue and in epithelial cells of the buccal mucosa, suggesting a partial overlap with the ARVC phenotype [34].

We investigated the molecular mechanisms of *FLNCtv* in the pathogenesis of ACM using patient-specific and genome-edited-induced pluripotent stem cell-derived cardiomyocytes (iPSC-CMs): we found that mutant *FLNCtv* models recapitulate the human phenotype, showing an arrhythmic profile and impaired contraction [35]. We also found that *FLNCtv* causes haploinsufficiency, leading to the activation of the platelet-derived growth factor receptor-α (PDGFRA) pathway. Indeed, the inhibition of PDGFRA with crenolanib was able to partially rescue *FLNCtv* iPSC-CMs’ contractility, suggesting a potential therapeutic strategy in FLNC-related ACM. More recent studies using iPSC-CMs models of *FLNCtv* have shown significant upregulation of focal adhesion signaling and dysregulation of thin filament genes in FLNC^KO−/−^ hiPSC-CMs compared to isogenic hiPSC-CMs, leading to cytoskeletal defects and the activation of focal adhesion kinase [35,36]. These data suggest that *FLNCtvs* induce complex alterations of the cardiomyocyte structure and function, and that heart failure and ventricular arrhythmias may be triggered not only by myocardial fibrosis but also by cell-intrinsic factors involving the derangement of the sarcomere, cytoskeletal, and intercalated disk structure, and calcium transient dysregulation [35,36].

In this current study, we applied a *mechanomic* approach to comprehensively investigate the interaction between altered biophysical and mechanical properties and biological response in hiPSC-CMs carrying *FLNCtv*. Furthermore, we investigated the effect of the inhibition of PDGFRA with crenolanib to assess its ability to rescue the altered biomechanics of mutant FLNC hiPSC-CMs, further supporting its potential therapeutic role.

## 2. Results and Discussion

### 2.1. FLNC Deletion Disrupts the Mechanical Properties of hiPSC-CMs

To investigate the mechanical effect of haploinsufficient FLNC in human cardiomyocytes, we used CRISPR/Cas9 genome-edited *FLNC^KO^*^−/−^ hiPSC-CMs, a homozygous condition which is lethal in animal models [27], and *FLNC^KO^*^+/−^, which represents the heterozygous human disease. The differentiation of hiPSC-CMs was performed using previously established procedures [37,38,39]. Wild-type *FLNC (FLNC^WT^)* hiPSC-CMs were used as controls. 

The biomechanical profile of hiPSC-CMs was investigated with atomic force microscopy. A JPK NanoWizard 4a BioScience AFM (JPK Instruments, Berlin, Germany) was used to acquire loading (approaching)/unloading (retraction) curves for single-cell force spectroscopy (SCFS). All studies were performed on living, intact cells in a cell culture medium, and only well-spread and isolated cells were investigated. Those that were clearly not adherent were rejected. In SCFS experiments, it is essential to assume that cells can be in different states and, thus, display distinct properties that are sometimes problematic to compare. To mitigate these difficulties, (i) multiple measurements from different cells were collected to control for variability and “average” data determined, and (ii) cells were monitored and their morphological details observed with optical light microscopy to allow for an appropriate selection of the cells throughout the tests. A cantilever coated with a layer of gold (CP-PNPL-Au-C, NanoAndMore GmbH, Wetzlar, Germany) modified with a 7 μm diameter polystyrene microsphere was used to precisely apply a physiological compression force to the cell in correspondence to the nuclear region. For soft biological samples such as cells, it is recommended to use spherical probes: in this way, the force is applied to a larger cell area causing a lower pressure and less cell damage compared to the sharp tip effects. Another reason is that spherical indenters average cell or tissue inhomogeneities, such as those originated by different components (nucleus, cytoskeletal components, etc.) (Figure 2A). Force–indentation curves recurrently display two different slopes (Figure 2B) and do not follow a simple power law function as predicted by the contact mechanic model: from these two different slopes, we were able to calculate E1 (Young modulus or elasticity/stiffness) as related to the compression of the outer cell’s superficial part (plasma membrane and actin cortex), and E2 which corresponds to the deeper cell structures (cytoskeleton and nucleus).

We indented 28 FLNC^WT^ cells from six distinct cultures as controls, 33 FLNC^KO+/−^ heterozygous hiPSC-CMs from three different cultures, and 28 FLNC^KO−/−^ homozygous hiPSC-CMs from three cultures. The same operators performed the experiments. For each cell, 20 measures were performed, and mean and standard deviation were calculated for each cell line. The test duration was never longer than 45–50 min to ensure cells’ viability. Using the JPK DP software, the approaching curves were transformed into force–indentation (F-D) curves by subtracting the cantilever bending from the signal height to calculate indentation(software URL: https://www.bruker.com/en/products-and-solutions/microscopes/bioafm.html; accessed on 28 October 2023). The F-D curves were then exported in .txt and fed to an Igor Pro^®^ (Wavemetrics Inc.) custom procedure to fit the data, thus calculating E.

To compare the mechanical properties of the different cells, the measurements were performed at a constant speed (1 µm/s), and the mathematical fits were carried out at a set indentation depth (500 nm): indeed, the calculated E has been found to differ appreciably on the specific choice of these parameters [40]. Thus, relative E variations were attained by keeping these parameters constant for all experiments. Furthermore, the speed range was chosen to avoid both cell movements (at a low compression speed) and a hydrodynamic force contribution (significant at a high speed).

Subsequently, the fit function described by the Hertz–Sneddon model was used [41]. Indeed, since the earliest AFM studies of soft biological samples, the prevalent method of evaluating AFM indentation data to assess the elasticity has been the “Hertz-Sneddon model” of contact between two elastic bodies [42]. The Hertz model assumes homogeneity, absolute cell elastic behavior, and no interactions between the sample and probe [43,44]. However, most biological materials are neither homogeneous nor completely elastic. In this case, the Hertz model is only valid for small indentations (up to 10% of the height of the cell). With the limitations that elasticity values calculated using various models differ from each other [45,46], the Hertz model is the preferred one for achieving information about cell elasticity.

In the present study, we used the Hertz–Sneddon model for spherical tips [41].
(1)$$F=\frac{4E\sqrt{R}}{3(1-\upsilon^{2})}\delta^{3/2}$$
where F is the load force, E is the Young modulus, ν the Poisson ratio, δ is the probe penetration into the cell. Poisson’s ratio was assumed to be 0.5. 

However, cells are not homogeneous, and deviations from this model are likely: the deformations of different structural layers will appear convoluted in the force–indentation curve, which can show the presence of different regimes. To take into account if different regimes exist and what relative influence they have on the elasticity, a modified Hertz model has been used [47]. A helpful visualization is an arrangement of stacked springs of increasing stiffness. As shown in (Figure 2A), an applied force would quickly deform the less stiff element. The stiffer element, however, would quickly snap back to its initial shape due to the higher Young’s modulus. Overall, we can assume that every single part behaves elastically and that the cell membrane layer is initially indented, followed by the cytoskeleton/nucleus part. Hence, the fitting curve can be written as follows [48]: (2)$$F\left(\delta \right)=\left\{\begin{array}{l} F_{0}\;\;\;\;\;\;\;\;\;\;\;\;\;if\;\delta >\delta_{1}\\ F_{0}+A1({\delta_{1}+\delta )}^{\frac{3}{2}}\;\;\;\;\;if\;\delta_{2}<\delta <\delta_{1}\\ F_{0}+A1{\left(\delta_{1}-\delta \right)}^{3/2}+A2{\left(\delta_{2}-\delta \right)}^{3/2}\;if\;\delta <\delta_{2} \end{array}\right.\;A1=\frac{4\;\;E1}{3\left(1-\nu^{2}\right)}\sqrt{R}\;A2=\frac{4\;\;E2}{3\left(1-\nu^{2}\right)}\sqrt{R}$$

F_0_ is the force at zero deformation, E1 and E2 are the Young modulus as obtained from the first and second curve slope, δ is the indentation, δ_1_ is the first point of contact between the cantilever and the cell, and δ_2_ is chosen as the position where the second slope becomes significant; δ_1_ and δ_2_ are assessed through the fitting procedure. 

When we investigated the hiPSC-CMs models (FLNC^WT^, FLNC^KO+/−^, and FLNC^KO−/−^), the force–indentation curve for the cells analyzed showed significant differences, as illustrated in Figure 2B, and as shown in the Young’s modulus assessment (Figure 3). Both elastic contributions (E1 and E2) showed that FLNC^WT^ hiPSC-CMs are stiffer than both mutant FLNC^KO+/−^ and FLNC^KO−/−^ hiPSC-CMs (Figure 3B,C). The stiffness significantly decreased in FLNC knockout iPSC-CM models in a gene dosage manner, as compared to control FLNC^WT^ hiPSC-CMs. As shown in Figure 3B,C, E1 Pa for FLNC^KO−/−^, FLNC^KO+/−^ and healthy control hiPSC-CMs were (E ± SD) 318 ± 356, 518 ± 314, and 1804 ± 1139, respectively. Interestingly, E2 Pa also significantly decreased in a gene dosage manner, as E2 Pa for FLNC^KO−/−^, FLNC^KO+/−^, and healthy control hiPSC-CMs were 1324 ± 1509, 3933 ± 5484, and 8219 ± 7619, respectively. The results of force indentation showed that the stiffness of hiPSC-CMs deficient of FLNC was significantly lower than the healthy control hiPSC-CMs, suggesting subcellular restructuring in hiPSC-CMs lacking FLNC (Table 1).

### 2.2. Loss of FLNC Also Reduces Adhesion Properties of hiPSC-CMs

The cell adhesion behavior was measured by assessing the retraction curve of the SCFS force–distance curves. The measured force/distance curves were processed in terms of adhesion force (the highest value of the force exerted by the cell-surface contact to the cantilever during retraction), rupture length (distance between cell surface and AFM tip at which the last connection breaks), and work of adhesion (integrating the area under the retraction curve), as shown in Figure 4A.

The adhesion cell behavior showed the same trend as the stiffness/elasticity (Figure 4B,C): compared to FLNC^WT^ iPSC-CMs, mutant FLNC iPSC-CMs showed progressive decrease in the work of adhesion as well as adhesion force from the FLNC^KO+/−^ to the FLNC^KO−/−^ models. Indeed, for the work of adhesion: FLNC^WT^ = 10.90 × 10^−16^ J, std.er. = 1.35; FLNC^KO+/−^ = 2.85 × 10^−16^ J, std.er. = 0.50; and FLNC^KO−/−^ = 0.95 × 10^−16^ J, std.er. = 0.13. For the adhesion force: FLNC^WT^ = 921.20 pN, std.er. = 64.25; FLNCKO+/− = 505.09 pN, std.er. = 39.87; and FLNC^KO−/−^ = 176.45 pN, std.er. = 14.77). Finally, the rupture length (Figure 4A) showed a similar trend and reached a statistical difference between FLNC^WT^ and the FLNC^KO−/−^ (*p* = 0.040) (Table 1).

### 2.3. Beating Frequency and Vertical Displacement in FLNC Models

Beating frequency (Figure 5) and the AFM tip’s vertical displacement showed no significant difference between the three groups. However, from a qualitative point of view, diastolic derangements were observed, especially in the case of the FLNC^KO−/−^ cells. Such alterations are shown in Figure 5 and are indicated by black arrows. They could be explained as temporary blockages of the cardiomyocytes’ contraction machinery during relaxation. We hypothesize that the loss of FLNC might contribute to such behavior by destabilizing the actin network and, thus, preventing a proper elastic rebound after contraction. Overall, 60% (6/10) FLNC^KO+/−^ and 83% (5/6) FLNC^KO+/−^ showed such derangements in the traces, compared to 10% (1/10) for the FLNC^WT^ control. Only FLNC^WT^ vs. FLNC^KO+/−^ reached statistical significance (*p* = 0.0012). This finding suggests that the derangement of cell structure and cytoskeletal network may cause a defective contraction and relaxation of the cardiomyocytes, which displaces the AFM tip vertically but does not translate in normal mechanical function. This irregular mechanical behavior recapitulates the arrhythmic profile observed in patients’ mutant FLNC iPS-CMs [35].

### 2.4. Defective Cell Adhesion in Mutant FLNC Is Associated with Altered Integrins Gene Expression

We have previously shown that mutant FLNC causes complex changes in human cardiomyocytes, involving the cytoskeleton, cell–cell junction, and signaling molecules. In particular, we found that desmosomal proteins (DSP, JUP) were downregulated. In addition, the expression level of connexin 43 and beta-catenin was unchanged, but their localization was altered: CX43 localized to the cytoskeleton and increased beta-catenin localization to the nucleus [35]. To further investigate if FLNC ablation induces similar molecular mechanisms that lead to cell–cell adhesion defects, we compared the transcriptome profiles of FLNC^KO+/−^ and FLNC^KO−/−^ with FLNC WT hiPSC-CMs. As shown in Figure 6, FLNC^KO+/−^ and FLNC^KO−/−^ cell lines showed altered integrins expression compared with the WT control. ITGA7, ITGB1BP2, and ITGB3 were downregulated in both the FLNC^KO+/−^ and FLNC^KO−/−^ models. Conversely, ITGB1, ITGB5, ITGA11, ITGA8, ITGA1, and ITGA2 were overexpressed in FLNC^KO+/−^ hiPSC-CMs when compared to FLNC^WT^. The homozygous FLNC^KO−/−^ hiPSC-CM model showed variability in expression. ITGA7, a receptor for the basement membrane protein laminin-1, has been shown to reduce adhesion to a substrate in aged muscle stem cells and is associated with congenital myopathy [49,50]. Finally, ILK (integrin-linked kinase) was approximately three-fold downregulated in FLNC^KO+/−^ (Fold Change: 2.81; *p* = 0.00164) and FLNC^KO−/−^ (Fold Change: 3.23; *p* = 0.0007) compared to FLNC^WT^ iPSC-CMs. The results might suggest that the dysregulation of the cell–ECM interactions reduced the adhesion properties of FLNC knockout hiPSC-CMs.

### 2.5. Recovering the Biomechanical Properties of FLNCtv Cardiomyocytes

Finally, we investigated if the defective mechanical properties of our *FLNCtv* cardiomyocyte models could be rescued by a therapeutic approach. For this purpose, we tested the mechanical effect of crenolanib, an inhibitor of the Platelet-Derived Growth Factor Receptor α (PDGFRA) pathway. In our previous investigations, we found that beta-catenin signaling and the downstream effector PDGFRA are upregulated in FLNC mutant hiPSC-CMs. When we treated cardiomyocytes with crenolanib, a PDGFRA inhibitor currently evaluated in clinical trials for various types of cancer, we observed partial normalization of the beta-catenin signaling pathway and recovery of the conduction velocity and the arhythmic phenotype of the FLNC hiPSC-CM mutant lines. Thus, in this study, we tested the hypothesis that crenolanib could also rescue the altered biomechanics of FLNC mutant hiPSC-CMs. We found that crenolanib-treated FLNC^KO−/−^ hiPSC-CMs partially rescued their stiffness (E2) and improved the adhesion properties (force and work of adhesion) compared to WT control (Figure 7 and Table 2). These findings further support the potential therapeutic role of crenolanib in FLNC cardiomyopathy.

In a recent study, Powers et al. [51] investigated neonatal mouse ventricular cardiomyocytes (NMVMs) with *FLNC* deletion using AFM. They found that in the FLNC knock-down NMVMs, the transverse indentation stiffness which indicates cortical tension was significantly lower than in controls. Based on these findings, which were complemented by computational models, the authors suggested that a reduction of FLNC could cause decreased Z-disk rigidity which, in turn, could alter the transmission of sarcomere forces and disrupt longitudinal force production during contraction.

Likewise, in our model of human cardiomyocytes, by combining AFM single-cell and molecular biology, we found that *FLNCtv* causes a decrease in cardiomyocyte stiffness, which declines in a gene dosage manner from the heterozygous (*FLNC^KO+/−^*), which has been shown to have a reduced amount of FLNC [31,35], to the homozygous (*FLNC^KO−/−^*), which lacks FLNC and has been found to be lethal in animal models [52]. Overall, these findings suggest a profound remodeling of structure and disruption of function in cardiomyocytes lacking FLNC.

It is also conceivable that both Young moduli (E1 and E2) diminish because FLNC interacts with cell membrane receptors offering a link between cytoskeletal actin and integrins in the membrane, essential for mechanotransduction and mechanosignaling. Furthermore, since FLNC provides cytoskeleton stiffness, creating actin networks, cells partially or completely lacking FLNC are much softer compared to controls. Another variable to be considered is the FLNC density. Indeed, filamins (FLNs) are ABPs that accomplish several assignments: besides actin crosslinkers, they connect the ECM with the actin network through integrins and play an important role in cell adhesion, signaling, and mechanotransduction. Another important characteristic of FLNs as ABP is that, due to their “V” shape, while being ineffective in forming strong bundles of actin filaments, they are able to form open actin networks [53]. This characteristic of FLNs, called angle-constraining crosslink, requires finite energy for rotating two crossing filaments with respect to each other. This contrasts with α-actinin which, instead, generates an energy-free-rotating crosslink of crossing filaments.

When cells sense stresses, they can go through changes in shape, spreading, and adhesion [54]. For these tasks, cells behave like a viscoelastic body, changing their mechanical properties to a more fluid-like behavior, when, for instance, they need to diffuse through tissues. In this regard, among the proteins making the physical structure of the cytoskeleton, actin is possibly the most important one. However, actin filaments are organized and coordinated by several actin-binding proteins. The main reason for this large number of ABD is that even though a single actin filament can produce piconewton forces, actin having a persistence length of 17.7 μm can easily buckle under load [55]. With a persistence length similar to its contour length, actin filaments are regarded as “semiflexible filaments”, frequently referred to as polymer chains with a high bending modulus. Actin needs, therefore, to be combined into networks to provide the needed cell mechanical stability. The simplest mechanism that provides some sort of mechanical property to a polymer network is entanglements, which intuitively result from steric interactions between polymer chains (actin in this case). At sufficiently high densities, actin chains confine each other’s motions from being snakelike into a narrow tube formed by contacts with the surrounding filaments. This comportment is modeled by the reptation theory [56]. However, entangled biopolymers can store elastic energy only on short timescales because at longer timescales, the filaments outflow the constrictions inflicted by entanglements. Long-term mechanical strength is, therefore, achievable only in the presence of long-term interactions, which can occur mainly by crosslinking, making ABP critical for the cell function. As a result, crosslinking leads to a more elastic response of the actin network by drastically reducing network rearrangements and actin filaments’ bending modes that can dissipate stress [57]. Furthermore, not only the type of ABP but also their density plays an important role [58]. For densely cross-linked networks, filament stretching controls deformation, causing a stress-stiffening regime and a high elastic modulus. As the cross-link density is reduced, filament bending rules and the networks’ stress weaken [30,31]. This behavior results from a change in the actin’s nematic orientation under stress. Moreover, if the cross-linking density is high, the activity of myosin molecular motors causes a cell’s elastic stiffening [59], while if the cross-linking density is low, cells can stress relax much faster, causing a more fluid-like behavior [60]. It is well known that the crosslinker density regulates, for the most part, the macroscopic network mechanics [61,62], and, therefore, any mutations affecting a crosslinker also change the network’s mechanical response. As a result, the stiffness (Young modulus) would also be related to the cross-linking density. To highlight the high efficiency of FLNs as ABD, in an in vitro work, actin crosslinked with FLNA was able to form a well-percolated network even at a moderately low crosslink density, so that the network was able to gel and act like an elastic entity in reaction to stress. Using a freely-rotating crosslink instead of FLNs, a much higher concentration would be necessary to increase network elasticity to a similar level [63]. While much of the current knowledge is based on FLNA, which shares 73% (https://www.uniprot.org/, accessed on 28 October 2023) identity of protein sequence with FLNC, it should be noted that data on FLNC are lacking. 

### 2.6. Defective Cell–Cell Adhesion in FLNCtv Cardiomyocytes

*FLNCtv* causes altered cell–cell structure and function in hiPSC-CMs [35]. We previously reported the altered localization of desmosomal proteins such as DSP and SAP95 in cardiac tissue of FLNCtv carriers, similar to the changes found in patients with arrhythmogenic right ventricular cardiomyopathy, a disease caused by desmosome gene mutations [31]. We have also shown that cell–cell-junction structures are dysregulated, including connexin-43, in hiPSC-CMs from patients with *FLNCtv* and genome-edited *FLNC^KO−/−^* [35]. Indeed, FLNC interacts with multiple cell adhesion proteins, including integrins [64].

In line with these data, in the current study, the biomechanical analysis of the work of adhesion and the force of adhesion showed a decrease in the adhesion properties from the heterozygous state (*FLNC^KO+/−^* ≈ 74% less adhesion work) to the homozygous state (*FLNC^KO−/−^* ≈ 91% less adhesion work). Indeed, the work of adhesion decreased by an order of magnitude (from 1 fJ in the WT to 0.095 fJ in the *FLNC^KO−/−^*). Likewise, the adhesion force decreased progressively from the WT to the *FLNC^KO−/−^* (≈45% and 81% less, respectively). Although the analysis of the hiPSC-CMs adhesion properties by AFM’s bare tip represents a nonspecific adhesion defect, significant evidence indicates a profound change in the structure of the cell–cell junction in *FLNCtv*. Indeed, the analysis of the integrin pathway expression in hiPSC-CMs of *FLNC^KO+/−^* and *FLNC^KO−/−^* showed a significant differential expression when compared with WT (Figure 6). Interestingly, similar dysregulation of the integrin pathway has been reported in another arrhythmogenic cardiomyopathy caused by *PKP2* mutations, suggesting a common mechanism of disrupted mechanosignaling and mechanotransduction [65].

FLNC has been found to interact with the β1A integrin subunit [66] and sarcoglycans [67] at the costamere and act as a link between myofibrils and sarcolemma. Indeed, FLNs are multifunctional proteins that have important roles in adhesion, migration, and mechanotransduction. FLNs bind to integrins, transmembrane receptors, that facilitate the signals and mechanical transduction between the extracellular matrix and the cytoskeleton. Cells need integrins for adhesion and migration, and their activation is synchronized by intracellular protein–protein interactions and phosphorylation. The role of FLN in binding to integrins is to create an integrin clustering and, therefore, a recruitment of actin filaments close to the cell membrane, enhancing the cell mechanical properties. In this regard, the ability of FLNs to cross-link actin filaments helps the stabilization of cell membrane integrins when stress is applied and reduces potentially dangerous Ca^2+^ fluxes [68]. FLNs have been shown to bind to β1A, β2, β3, and β7, using domain 21, and, to a less extent, domain 19 [69,70,71,72,73]. Small variations of the interacting residues may modify the affinity of the integrin–filamin interaction that is mainly due to the hydrogen bonding and hydrophobic interactions. These changes include the sequence variations in integrin tails [74] as well as changes in binding residues in FLN 19 or other domains [69]. There are several examples of the influence of FLNs on cell adhesion. The siRNA knockdown of FLNA leads to a decrease of around 50% in both cell adhesion and in the formation of filopodia and lamellipodia during cell spreading [75]. FLNA-lacking cells display a reduced membrane expression of β1 integrins [76]. Together, these results imply that FLNs regulate key cell adhesion proteins of the cell membrane and, possibly, the molecules that affect their activity.

### 2.7. Inhibition of PDGRA Upregulation Improves Biomechanical Properties of FLNCtv Cardiomyocytes

In our previous investigations, we found that the loss of FLNC altered beta-catenin localization, activating the PDGFRA signaling pathway, which led to mislocalization of connexin-43 [35]. We also found that crenolanib, a PDGFRA inhibitor, could partially rescue the beta-catenin pathway pathological signaling, contractility, and electrical instability in patient-specific *FLNCtv* hiPSC-CMs. Thus, here, we investigated the cell biomechanical effects of crenolanib in our human cardiomyocyte models. We found that crenolanib was able to partially rescue the stiffness of FLNC^KO−/−^ hiPSC-CMs compared to control. Moreover, both adhesion force and the work of adhesion showed significant recovery with respect to control, as well. These mechanical data further support its potential therapeutic role in *filaminopathy* treatments.

It should also be noted that PDGFRA is known to act through receptor tyrosine kinases, which are regulators of proliferation, migration (in embryogenesis), metabolism, but also cellular proinflammatory stress in cardiomyocytes. At the same time, it is now known that various types of arrhythmias can be associated with mechanisms of cellular stress, such as hypoxia, lipotoxic factors, cellular ageing, dysfunction of inotropic, and metabotropic cardiomyocyte receptors. If the cytoskeletal changes observed are consequences of the pathogenetic role of cellular stress and inflammatory factors in FLNC dysfunction, or of the specific genetic defect, FLNC is still undefined, and more extensive research is needed [77].

## 3. Materials and Methods

### 3.1. FLNC^KO^ hiPSC-CM Models

Differentiation of hiPSC-CMs was performed as previously described [37,38,39]. Details of the differentiation of the iPSC lines FLNC^KO+/−^ and FLNC^KO−/−^ into cardiomyocytes have been previously described in detail in Chen et al., *Sci Adv.* 2022 [35], and Gao et al., *Cells* 2024 [36]. For genome-editing of hiPSC-CM models using the CRISPR/Cas9 system, a ribonucleoprotein complex (RNP) complex was formed in vitro using CAS9 protein, gRNA, and tracer RNA purchased from Integrated DNA Technologies, Inc. (Coralville, IA, USA). The RNP complex was electroporated using Amaxa Nucleofector (Lonza Bioscience, Morrisville, NC, USA) into hiPSCs lines isolated from single-cell colonies. Briefly, single hiPSCs were resuspended in a mixture containing the RNP and nucleofection^®^ followed by electroporation using program B23 of AMXA machine (www.amaxa.com, Cologne, Germany, accessed on 21 February 2024). Detection of mutated lines was performed using Sanger sequencing, confirming the heterozygous FLNC^KO+/−^ (c.549_550insT) and homozygous FLNC^KO−/−^ (c.549_554delTGACTG/c.549_550insT) [35]. hiPSCs were cultured in standard conditions using mTeSR1 media (STEMCELL Technologies, Vancouver, BC, Canada) and the feeder-free substrate Matrigel (Corning, NY, USA). Cardiac maturity was determined using cardiac markers (upregulated MYH7, HCN1, RYR2; downregulated GATA4, MEF2C, NKX2, and NKX2-5) [78].

### 3.2. Atomic Force Microscopy—Single Cell Force Spectroscopy

A JPK NanoWizard 4a BioScience AFM (JPK Instruments, Berlin, Germany) was used to acquire loading (approaching)/unloading (retraction) curves for single-cell force spectroscopy (SCFS) tests. Performing SCFS tests, (i) multiple measurements from different cells were collected to control for variability and “average” data determined, and (ii) cells were monitored and their morphological details observed (an optical light microscope was used for cell selection throughout the tests). The AFM was equipped with a PetriDishHeater™ tool to operate in liquid at a controlled temperature (37 °C). A gold-coated cantilever (CP-PNPL-Au-C, NanoAndMore GmbH, Wetzlar, Germany) modified with a 7 μm-diameter polystyrene microsphere was used to precisely apply a compression force “normal” to the nucleus. Polystyrene bead is not a truly rigid body: however, polystyrene’s Young modulus is about 3 × 10^9^ Pa much higher than that we measured for our nuclei (around 1 × 10^3^ Pa). AFM probes were cleaned, prior to the indentation experiments, by embedding them successively in Tween (2% for 30 min) to remove contaminant molecules adsorbed on the probe surface. All studies were performed on living, intact cells in cell culture medium. Only well-spread healthy and isolated cells were investigated. 

We assessed the cell elasticity using 33 heterozygous FLNC^KO+/−^ iPSC-CMs (from 3 cultures in 3 different days), 28 WT iPSC-CMs (6 cultures, 3 days), and 28 homozygous FLNC^KO−/−^ (2 different cultures, 2 days) cells for each condition, and the same operators performed the experiments. These data had the power to detect statistically significant differences. For each cell, 20 measures were performed, and mean and SD were calculated for each cell line.

The test duration was never longer than 45–50 min to ensure the cells’ viability. Cell stiffness/elasticity was obtained by evaluating the Young modulus (E) of the cell. It was calculated by analyzing the approaching part of the recorded “loading” curves using the JPK DP software. The software transforms the approaching curve into force–indentation curves by subtracting the cantilever bending from the signal height to calculate indentation. Subsequently, the fit function described by the Hertz–Sneddon model was used [41].

The cell adhesion behavior was measured by assessing the retraction curve of the SCFS force–distance curves. The measured force/distance curves were computed in terms of adhesion force (the maximal value of the force exerted by the cell-surface contact to the cantilever during retraction), rupture length (distance between cell surface and AFM tip at which the last connection breaks), and work of adhesion (integrating the area under the retraction curve), as shown in Figure 4. Beating frequency and vertical displacement were obtained from at least a 15 s recording in 9 FLNC^WT^, 11 FLNC^KO+/−^, and 5 FLNC^KO−/−^ beating iPSC-CMs (see Appendix A).

### 3.3. Statistical Analysis

For each of the cellular phenotyping experiments, we generated data from ≥3 separate biological replicates for each sample (Cas9 edited, control). ANOVA was used to evaluate differences between groups and false discovery rate adjustments to account for multiple comparisons. For AFM quantification of elasticity (Young modulus), cytoskeletal viscosity, and cell adhesion, data were derived from the force deformation loading/unloading curves as described above, and they are based on our previous work [79,80,81]. WT hiPSC-CMs, free of any known ACM mutations, provided control data. Means, medians, and SDs were calculated. Repeated measures data analyses were performed using linear mixed-effects models [82] and analyzed using open source R software(Version 4.2.3) [83]. The data were plotted, and if they appeared to be normal, ANOVA was performed. If there were major deviations from normality, in particular, large values of skewness and kurtosis, then Kruskal–Wallis was used. For multiple comparisons, Dunn’s test with statistical hypothesis-testing correction was performed. Hence, adjusted *p*-values are reported.

## 4. Conclusions

In this study, we used AFM-SCFS micro-indentation to evaluate the passive and dynamic mechanical properties of human cardiomyocytes carrying *FLNCtv* and causing a severe form of cardiomyopathy. We observed that beating traces showed irregular peak profiles in mutant FLNC^KO+/−^ and FLNC^KO−/−^ hiPSC-CMs, suggesting an “arrhythmic” behavior. Membrane (E1) and cell (E2) stiffness measured using the Young modulus showed a progressive decrease from FLNC^WT^ to mutant FLNC^KO+/−^ and FLNC^KO−/−^ iPSC-CMs. In addition, mutant FLNC iPSC-CMs showed a progressive decrease in work of adhesion and adhesion force from the heterozygous FLNC^KO+/−^ to the FLNC^KO−/−^ model compared to the FLNC^WT^ control, suggesting altered cytoskeleton and membrane structures. Finally, we investigated the biomechanical effect of crenolanib, an inhibitor of PDGFRA, which is upregulated in FLNCtv hiPSC-CMs, and found that crenolanib is able to partially rescue the stiffness and adhesive membrane properties of mutant FLNCtv cardiomyocytes, supporting the potential therapeutic applications of PDGFRA inhibitors.

## Figures and Tables

**Figure 1 ijms-25-02942-f001:**
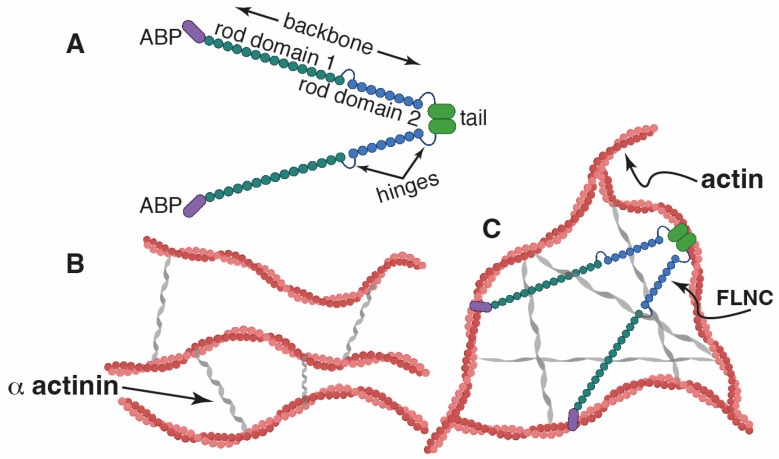
**Cartoon showing the predicted filamin structure (FLNC).** (**A**) Structure of FLNC. (**B**,**C**) Predicted interaction of filamin C, which shares a high degree of homology with (FLNA), with actin. (**B**) Without FLNC, only bundles of actin filaments are formed. (**C**) When FLNC is present, open actin networks are formed. The crucial role of FLNs is their aptitude to alter a population of fluid-like actin filaments into a network structure, consistently increasing the structural rigidity. Figure created with BioRender.com.

**Figure 2 ijms-25-02942-f002:**
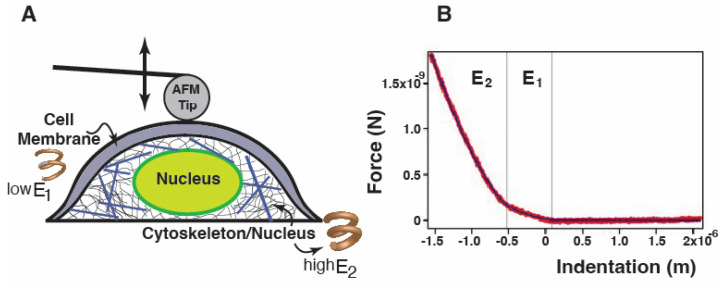
**hiPSC-CMs’ single-cell spectroscopy.** (**A**) Schematic representation of the AFM methodology used to test the stiffness of cardiomyocytes using the spheric tip. (**B**) Scheme of the two distinct components of the indentation curve, which include E1 and E2, where E1 is related to the compression of the outer cell superficial part. E1 is the Young modulus, and since the cell membrane has a lower Young’s modulus than the deeper cell part, E1 < E2. E2, the deeper components of the cell (cytoskeleton and nucleus), behaves as a stiffer spring compared to E1.

**Figure 3 ijms-25-02942-f003:**
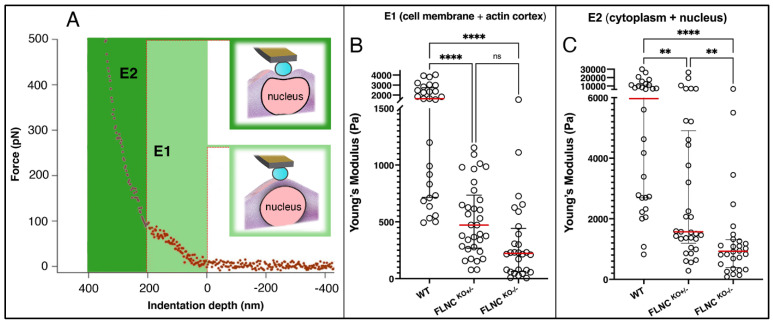
**FLNC^KO^ cardiomyocytes are softer.** Two elastic regimes (light- and dark-green-highlighted regions) are distinguishable from the AFM indentation curve (**A**). Young’s moduli (YM) are computed through least-square fitting to a two-component Hertz model with Sneddon modification (equation in main text). E1 represents the YM relative to the cell membrane and actin cortex (**B**). E2 represents the YM relative to the underlying cytoplasmic structures (e.g., the nucleus and inner actin network, **C**). For E1 (**B**), heterozygous FLNC^KO+/−^ CMs are softer than WT (*p* < 0.0001). Homozygous FLNC^KO−/−^ CMs are softer than WT (*p* < 0.0001) and FLNC^−/+^ as well but without statistical relevance (*p* = 0.061). For E2 (**C**), heterozygous FLNC^KO+/−^ CMs are softer than WT (*p* = 0.0085). Homozygous FLNC^KO−/−^ CMs are softer than WT (*p* < 0.0001) and FLNC^−/+^ as well (*p* = 0.0060). ** *p* < 0.01, **** *p* < 0.0001.

**Figure 4 ijms-25-02942-f004:**
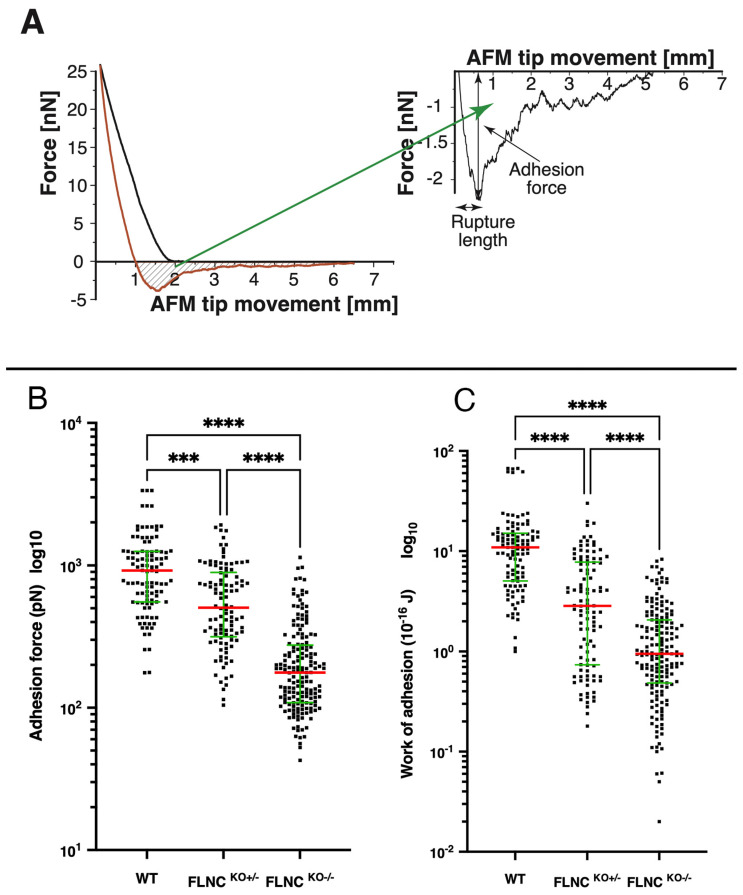
**FLNC^ko^ cardiomyocytes showing progressive loss of adhesion.** As shown in panel (**A**), adhesion force (**B**) and work of adhesion (**C**) are computed as the maximum detachment force and the retraction-baseline area of the retraction curve (green highlighted region), respectively. FLNC^−/+^ are less adhesive to the AFM tip than WT (*p* = 0.0002). FLNC^−/−^ further lose membrane adhesive properties with respect to WT (*p* < 0.0001) and to FLNC^−/+^ (*p* < 0.0001). Note that, in this case, each graphed dot represents an individual measurement. *** *p* < 0.001; **** *p* < 0.0001.

**Figure 5 ijms-25-02942-f005:**
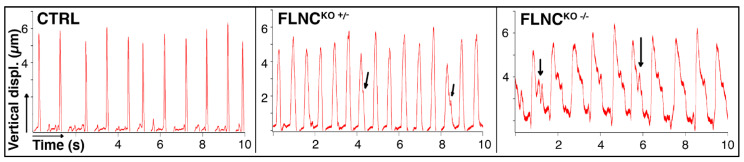
**Mutant FLNC disrupts the *diastolic* relaxation phase in hiPSC-CMs.** Three representative mechanocardiograms (AFM tip’s vertical displacement vs. time) showing progressive delay during the relaxation cycle, the severity of such being correlated with the severity of the mutation. Black arrows point out such delays during the cardiomyocyte’s relaxation phase. Mechanically, this could be the result of a temporary blockage in the CM contraction machinery due to the truncation of FLNC.

**Figure 6 ijms-25-02942-f006:**
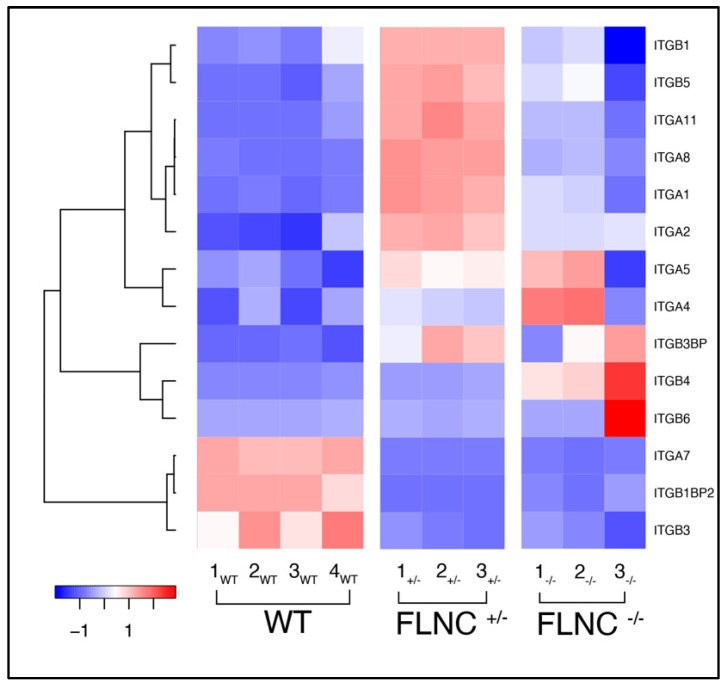
**Integrins expression profile in mutant FLNC iPSC-CM.** FLNC^KO+/−^ and FLNC^KO−/−^ cell lines showing altered integrins expression with compared to control WT iPSC-CM. ITGA7, ITGB1BP2, and ITGB3 are downregulated in both FLNC^+/−^ and FLNC^KO−/−^ models. The remaining integrins (from ITGB1 to ITGB3BP) are markedly overexpressed in FLNC^KO+/−^, while being mildly overexpressed or slightly downregulated in the case of FLNC^KO−/−^, when compared to WT.

**Figure 7 ijms-25-02942-f007:**
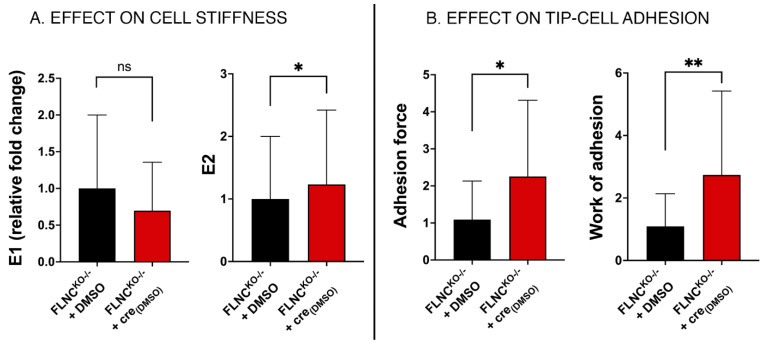
**Crenolanib-mediated mechanical rescue.** PDGFRA cascade inhibition induces a moderate stiffness recovery at the cytoplasmic regime, compared to untreated FLNC^KO−/−^ (*p* = 0.031), as shown by E2 distribution (panel (**A**)). The cell membrane and actin cortex do not stiffen, as shown by E1 (panel (**A**)). Cell-AFM tip adhesion shows moderate recovery in the case of Crenolanib (*cre* in graphs)-treated FLNC^KO−/−^ for both adhesion force (*p* = 0.0131) and work of adhesion (*p* = 0.0088) as shown in panel (**B**). * *p* < 0.05; ** *p* < 0.01.

**Table 1 ijms-25-02942-t001:** Young modulus and adhesion forces in hiPSC-CM models with homozygous (FLNC^KO−/−^) and heterozygous (FLNC^KO+/−^) FLNC truncation variants and wild type (FLNC^WT^) control.

MEAN ± SD				
Cells	E1 (Pa)	E2 (Pa)	Max Adh Force (pN)	Work of Adhesion (10^−16^ J)
FLNC^WT^, N = 28	1804 ± 1139	8219 ± 7619	1036 ± 659	13.4 ± 13.9
FLNC^KO+/−^, N = 33	518 ± 314	3933 ± 5484	233 ± 194	4.77 ± 5.17
FLNC^KO−/−^, N = 28	318 ± 356	1324 ± 1509	630 ± 411	1.57 ± 1.64
**MEDIAN/IQR**				
**Cells**	**E1 (Pa)**	**E2 (Pa)**	**Max Adh Force**	**Work of Adhesion**
WT N = 28	1637/2020	5966/8536	921/699	10.9/10.0
FLNC +/− N = 33	471/467	1570/3720	505/577	2.85/7.07
FLNC −/− N = 28	222/379	927/916	177/167	0.949/1.58

**Table 2 ijms-25-02942-t002:** Young modulus (E2) and adhesion forces in hiPSC-CM models with homozygous (FLNC^KO−/−^) FLNC truncation variants and wild type (FLNC^WT^) control treated with crenolanib.

MEAN ± SD			
Cells	E2	Max Adh Force (pN)	Work of Adhesion (0.1 fJ)
CTRL N = 22	1048 ± 894	95 ± 56	0.244 ± 0.259
CRE N = 22	1257 ± 1066	165 ± 116	0.696 ± 0696
**MEDIAN/IQR**			
**Cells**	**E2**	**Max Adh Force**	**Work of Adhesion**
CTRL N = 22	755/409	77/60	0.140/0.259
CRE N = 22	984/427	110/187	0.260/1.19

## Data Availability

Data is contained within the article and Appendix A.

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
