# Peer review of "Defective Biomechanics and Pharmacological Rescue of Human Cardiomyocytes with Filamin C Truncations"

_ijms, 2024, doi:10.3390/ijms25052942_

Round 1

Reviewer 1 Report

Comments and Suggestions for Authors

In the original article 'Defective Biomechanics and Pharmacological Rescue of Human Cardiomyocytes with Filamin C Trucations', submitted by Lazzarion and coworkers to the IJMS, the authors analyzed the biomechanical properties of iPSC-derived cardiomyocytes carrying FLNC truncation variants.

The topic of this article is highly interesting and the manuscript is well written. Nevertheless, I would change some points:

1.) Within the introduction I would explain that the cardiac intermediate filaments consists mainly of the intermediate filament protein desmin (see 'Molecular insights into cardiomyopathies associated with desmin (DES) mutations').

2.) In addition to the described truncating variants there are several missense mutations in FLNC known which cause an aberrant protein aggregation (p.S1624L and p.I2160F). It seems that truncating variants are more associated with DCM and missense mutations are more prominent in restrictive cardiomyopathy (RCM). I would mention pathogenic missense mutations leading to abnormal protein aggregation in RCM patients.

3.) Could you define the specific trucating variants in FLNC which you introduced? Which position? Which exon? 

4.) The altered integrin expression is interesting. However, I am wondering about ILK (integrin linked kinase). Is this adapter molecule also differentially expressed?

5.)Line 451: Could you explain the experimental conditions used for electroporation.

6.) In general, it would be necessary to explain the differentiation of iPSC into cardiomyocytes in more detail.

7.) Line 502: was performed instead of 'will be performed'. Please change.

In summary, the study is interesting and deserves publication. However, there are some points which should be changed. Therefore, I suggest a major revision. However, I am optimistic that the authors can fix the specific points in a revision. Good luck!

Comments on the Quality of English Language

Some minor corrections might be necessary. 

Author Response

We appreciate the positive comments of the Reviewers. Below, we reply to the Reviewers’ comments/criticisms in a point-by-point-manner: modifications in the manuscript are highlighted.

Reviewer #1.

  1. Within the introduction I would explain that the cardiac intermediate filaments consist mainly of the intermediate filament protein desmin (see 'Molecular insights into cardiomyopathies associated with desmin (DES) mutations').

A sentence and a reference (Brodhel et al., 2018) have been added to the Introduction (line 44 - 45), as suggested.

  1. In addition to the described truncating variants there are several missense mutations in FLNC known which cause an aberrant protein aggregation (p.S1624L and p.I2160F). It seems that truncating variants are more associated with DCM and missense mutations are more prominent in restrictive cardiomyopathy (RCM). I would mention pathogenic missense mutations leading to abnormal protein aggregation in RCM patients.

A sentence has been added to the Introduction regarding abnormal protein aggregation in RCM as suggested by the Reviewer (lines 91 - 94).

  1. Could you define the specific truncating variants in FLNC which you introduced? Which position? Which exon? 

Details of the truncating variant were previously described (Chen et al., Sciences Advances 2022). For clarity, we have added the details in the manuscript: Materials and Methods, lines 131-145.

  1. The altered integrin expression is interesting. However, I am wondering about ILK (integrin linked kinase). Is this adapter molecule also differentially expressed?

ILK was approximately 3-fold downregulated in FLNCKO+/- (Fold Change: 2.81; p=0.00164) and FLNCKO-/- (Fold Change: 3.23; p=0.0007) compared to healthy iPSC-CMs. This data has been added to the Results, lines 372 - 375.

  1. Line 451: Could you explain the experimental conditions used for electroporation.

Details added in Materials and Methods, lines 134-140.

Briefly, single hiPSCs were resuspended in a mixture containing the RNP and nu-cleofection® followed by electroporation using program B23 of AMXA machine (www.amaxa.com, Germany). This information has been added to the Materials and Methods section. However, the program detail is a proprietary property of AMAXA (chrome-extension://efaidnbmnnnibpcajpcglclefindmkaj/https://timothyspringer.org/files/tas/files/amaxa_nucleofectorii.pdf). 

  1. In general, it would be necessary to explain the differentiation of iPSC into cardiomyocytes in more detail.

Methods for the differentiations of iPSC into cardiomyocytes have been previously reported in details in: 1. Chen, Lam et al., Activation of PDGFRA signaling contributes to filamin C-related arrhythmogenic cardiomyopathy. Sci Adv. 2022 Feb 25;8(8):eabk0052. 2. Gao et al. Filamin C Deficiency Impairs Sarcomere Stability and Activates Focal Adhesion Kinase through PDGFRA Signaling in Induced Pluripotent Stem Cell-Derived Cardiomyocytes. Cells. 2024 Feb 2;13(3):278. doi: 10.3390/cells13030278. PMID: 38334670; PMCID: PMC10854597.  This has been clarified in the Materials and Methods, lines 131-145.

  1. Line 502: was performed instead of 'will be performed'. Please change.

Corrected as indicated (line 549).

Reviewer 2 Report

Comments and Suggestions for Authors

The subject of the article is relevant. Understanding the events that take place in normal and diseased heart cells is crucial, as they are important determinants of overall heart function. In addition to chemical and molecular events, there are structural and mechanical phenomena that need to be studied. Cell structure and mechanics are largely dependent on the cytoskeleton, which is composed of filamentous proteins that can be cross-linked by accessory proteins. Alpha-actinin 2 (ACTN2), filamin C (FLNC) and dystrophin are three major actin cross-linkers that are extensively involved in the regulation of cell structure and mechanics. The article makes a good impression, including the problem formulated, the quality of the results obtained and the research methods used. I have no comments of a principled nature on this work, but there are two comments of an unprincipled nature:

1. Authors investigated the biomechanical effect of crenolanib, an inhibitor of PDGFRA which is upregulated in FLNCtv, and found that crenolanib is able to partially rescue the stiffness and adhesive membrane properties of mutant FLNCtv cardiomyocytes, supporting the potential therapeutic applications of PDGFRA inhibitors. At the same time, PDGFR is known to act through receptor tyrosine kinases, which are regulators of proliferation, migration (in embryogenesis), metabolism, but also cellular proinflammatory stress in cardiomyocytes. The signalling pathways of receptor tyrosine kinases (including PDGFRA) largely overlap and interact with the signalling pathways of various metabotropic neurotransmitter receptors acting through G-proteins. At the same time, it is now known that various types of arrhythmia can be associated with typical mechanisms of cellular stress, in particular those induced by hypoxia, lipotoxic factors, cellular ageing, dysfunction of inotropic and metabotropic cardiomyocyte receptors. From this point of view, it would be interesting to know what the authors think about the possible universality of the patterns of cytoskeletal changes they have identified, namely as a typical pathogenetic mechanism in different types of cardiomyopathies, or whether additional, more extensive research is needed? The first is the pathogenetic role of cellular stress and inflammatory factors in filamin C dysfunction, or are the patterns revealed only limited by genetic mutations? To answer these questions, perhaps the authors of the article should familiarise themselves with the recent work of Wang and co-authors (https://doi.org/10.1101/2024.01.05.574393).

2. Figure 1(C). FLNС is shown, FLNA is written in the note.

Author Response

Reviewer #2

  1. Authors investigated the biomechanical effect of crenolanib, an inhibitor of PDGFRA which is upregulated in FLNCtv, and found that crenolanib is able to partially rescue the stiffness and adhesive membrane properties of mutant FLNCtv cardiomyocytes, supporting the potential therapeutic applications of PDGFRA inhibitors. At the same time, PDGFR is known to act through receptor tyrosine kinases, which are regulators of proliferation, migration (in embryogenesis), metabolism, but also cellular proinflammatory stress in cardiomyocytes. The signalling pathways of receptor tyrosine kinases (including PDGFRA) largely overlap and interact with the signalling pathways of various metabotropic neurotransmitter receptors acting through G-proteins. At the same time, it is now known that various types of arrhythmia can be associated with typical mechanisms of cellular stress, in particular those induced by hypoxia, lipotoxic factors, cellular ageing, dysfunction of inotropic and metabotropic cardiomyocyte receptors. From this point of view, it would be interesting to know what the authors think about the possible universality of the patterns of cytoskeletal changes they have identified, namely as a typical pathogenetic mechanism in different types of cardiomyopathies, or whether additional, more extensive research is needed? The first is the pathogenetic role of cellular stress and inflammatory factors in filamin C dysfunction, or are the patterns revealed only limited by genetic mutations? To answer these questions, perhaps the authors of the article should familiarise themselves with the recent work of Wang and co-authors (https://doi.org/10.1101/2024.01.05.574393).

We are grateful to the reviewer for these important considerations. A paragraph discussing the possible pathogenetic mechanisms of cytoskeletal changes and arrhythmias and a citation of the work of Wang et al. have been added, lines 528-535.

  1. Figure 1(C). FLNС is shown, FLNA is written in the note.

Figure 1 legend has been corrected and clarified.

Reviewer 3 Report

Comments and Suggestions for Authors

The article is very interesting and the data obtained by the authors are valuable for both science and biomedicine as they provide a better understanding of the disease mechanism associated with truncated filamin C variants. The article is also interesting from a methodological point of view, as it uses a combination of approaches such as CRISPR/Cas-mediated genome editing, atomic force microscopy and transcriptomic analysis to study in depth the molecular mechanisms of the disease. Nevertheless, the article contains certain shortcomings that should be corrected.

1.       The filamin C truncated variants (FLNCtv) are mentioned in the article. I think it would be very valuable to give at least a brief overview of what exactly these mutations are, how common they are in the population, and whether their manifestations differ phenotypically, including in terms of the interpretability of the results obtained.

2.       The Cas9 protein, gRNA, and tracrRNA should be described in detail in the Materials and Methods section. The sequence of gRNA is particularly important because it determines the location in the genome that will be cleaved by the editor, so it should be indicated.

3.       It is necessary to compare the genotype of the mutant cells obtained in the work with existing variants of filamin C truncation (FLNCtv). This is essential for proper interpretation of the results and evaluation of their applicability to different patients.

4.       Evidence that the mutant cells are indeed homo- or heterozygous filamin C knockouts should be provided. This can be done by Sanger sequencing of this genomic region and added to the article as a supplementary file or an additional figure panel.

5.       Since transcriptome data have already been obtained, a differential gene expression and functional analysis (GO and KEGG terms) would be highly desirable. This would allow us to assess the impact of mutations at the expression level of all genes, not just individually selected ones.

6.       The caption to Figure 1 mentions filamin A, but the illustration itself only contains filamin C. This should be corrected.

7.       The arrangement of the captions in Figure 2 is somewhat confusing. It is difficult to understand what "high" and "low" correspond to, and can they be moved away from "E2" and "E1", respectively? The caption "tip" should be moved closer to the tip itself, and the caption "AFM" should be moved to the left, away from the arrow indicating the oscillation of the cantilever, if I understand the meaning of the figure correctly.

8.       The caption to Figure 3 indicates "*p<0.05, ***p<0.0001". It is necessary to specify which p-values correspond to ** and ****.

9.       In Figure 5, the mechanocardiograms show additional smaller peaks. They are marked with arrows, and it is stated that "black arrows indicate such delays during the relaxation phase of cardiomyocytes". It is clear that in this case we are talking about temporary delays, but can we somehow justify the presence of the teeth themselves?

10.   When comparing the 3 groups (WT, KO+/-, KO-/-) on certain parameters (e.g. E1, E2, adhesion force, work of adhesion) were multiple comparison corrections introduced? If no, this is worth doing, if yes, it should be indicated and write which correction was used, as this may affect the p-value.

11.   This is a formal requirement, but the sections need to be reversed somewhat, and the sections and subsections need to be enumerated as in the MDPI template.

Comments on the Quality of English Language

English is good

Author Response

Reviewer #3

  1. The filamin C truncated variants (FLNCtv) are mentioned in the article. I think it would be very valuable to give at least a brief overview of what exactly these mutations are, how common they are in the population, and whether their manifestations differ phenotypically, including in terms of the interpretability of the results obtained.

We are grateful to the reviewer for this important comment: indeed, a discussion on the relevance of FLNC truncations in the patient populations has been added, lines 98-107.

  1. The Cas9 protein, gRNA, and tracrRNA should be described in detail in the Materials and Methods section. The sequence of gRNA is particularly important because it determines the location in the genome that will be cleaved by the editor, so it should be indicated.

As reported in the Reviewer #1 point 3, details of the methods for generating the truncating variant were previously described (Chen et al., Sciences Advances 2022; Gao et al., Cells 2024). The specific mutations are now reported in Material and Methods. The images of the sequences for the FLNCKO-/- and FLNCKO+/- were previously reported in Chen et al., Sciences Advances 2022, Supplementary Materials, Figure S3. For clarity, we have added more details in the manuscript: Materials and Methods, lines 131-145.

  1. It is necessary to compare the genotype of the mutant cells obtained in the work with existing variants of filamin C truncation (FLNCtv). This is essential for proper interpretation of the results and evaluation of their applicability to different patients.

A discussion on the comparison of the genome-edited FLNCtv with existing variants has been added to the discussion (see above point #1), line 98-107. Furthermore, FLNCtv are also discussed in lines 198-201, 472-479.

  1. Evidence that the mutant cells are indeed homo- or heterozygous filamin C knockouts should be provided. This can be done by Sanger sequencing of this genomic region and added to the article as a supplementary file or an additional figure panel.

Please see above point #2. Furthermore, we add below for the Reviewer the images of the sequences, from Chen et al., Sciences Advances 2022, Supplementary Materials, Figure S3.

Fig. S3. Pluripotent stem cell markers and genotypes of FLNC knock out iPSC lines. (A) Immunofluorescence staining against pluripotent stem cell markers of FLNC knockout iPSC lines. (B) Sanger sequencing confirmation of FLNCKO+/- and FLNCKO-/- knockout.

  1. Since transcriptome data have already been obtained, a differential gene expression and functional analysis (GO and KEGG terms) would be highly desirable. This would allow us to assess the impact of mutations at the expression level of all genes, not just individually selected ones. 

We appreciate the comment of the Reviewer: the extensive investigation of all differentially expressed genes is currently ongoing and has been, in part, recently published in a separate paper (Gao et al. Filamin C Deficiency Impairs Sarcomere Stability and Activates Focal Adhesion Kinase through PDGFRA Signaling in Induced Pluripotent Stem Cell-Derived

Cardiomyocytes. Cells. 2024 Feb 2;13(3):278. doi: 10.3390/cells13030278. PMID:

38334670; PMCID: PMC10854597).

            A discussion of these important aspect has now been added to the Introduction (see above, point #1), lines 115-122.

  1. The caption to Figure 1 mentions filamin A, but the illustration itself only contains filamin C. This should be corrected.

Figure 1 legend has been corrected and clarified, see also Reviewer #2 point 2.

  1. The arrangement of the captions in Figure 2 is somewhat confusing. It is difficult to understand what "high" and "low" correspond to, and can they be moved away from "E2" and "E1", respectively? The caption "tip" should be moved closer to the tip itself, and the caption "AFM" should be moved to the left, away from the arrow indicating the oscillation of the cantilever, if I understand the meaning of the figure correctly.

Done. We hope that now the figure and caption are clarified.

  1. The caption to Figure 3 indicates "*p<0.05, ***p<0.0001". It is necessary to specify which p-values correspond to ** and ****.

Corrected: ****p<0.0001, and added ***p<0.001 and **p<0.01

  1. In Figure 5, the mechanocardiograms show additional smaller peaks. They are marked with arrows, and it is stated that "black arrows indicate such delays during the relaxation phase of cardiomyocytes". It is clear that in this case we are talking about temporary delays, but can we somehow justify the presence of the teeth themselves?

For clarity, we added the following (lines 340-343): “They could be explained as temporary blockages of the cardiomyocytes contraction machinery during relaxation. We hypothesize the loss of FLNC might contribute to such behavior by destabilizing the actin network and thus preventing a proper elastic rebound after contraction.”

  1. When comparing the 3 groups (WT, KO+/-, KO-/-) on certain parameters (e.g. E1, E2, adhesion force, work of adhesion) were multiple comparison corrections introduced? If no, this is worth doing, if yes, it should be indicated and write which correction was used, as this may affect the p-value.

For multiple comparisons, Dunn’s test with statistical hypothesis testing correction was used (and adjusted p-values are therefore reported). We added the following sentence at the end of the statistical analysis section to clarify that: “For multiple comparisons, Dunn’s test with statistical hypothesis testing correction was performed. Hence, adjusted p-values are reported.” (lines 193-194)

  1. This is a formal requirement, but the sections need to be reversed somewhat, and the sections and subsections need to be enumerated as in the MDPI template.

Done.

Round 2

Reviewer 1 Report

Comments and Suggestions for Authors

Congratulations! The authors have improved their manuscript. I suggest to accept this manuscript for publication in its current form! 

Reviewer 3 Report

Comments and Suggestions for Authors

The authors have substantially improved the manuscript by responding to all reviewers' comments. I have no further significant comments and recommend the revised manuscript for publication in the journal.